# Beneficial Effects of Saw Palmetto Fruit Extract on Urinary Symptoms in Japanese Female Subjects by a Multicenter, Randomized, Double-Blind, Placebo-Controlled Study

**DOI:** 10.3390/nu14061190

**Published:** 2022-03-11

**Authors:** Shizuo Yamada, Michiyo Shirai, Ken Ono, Shinji Kageyama

**Affiliations:** 1Center for Pharma-Food Research, Graduate School of Pharmaceutical Sciences, University of Shizuoka, Shizuoka 422-8526, Japan; cpfr02@u-shizuoka-ken.ac.jp; 2Izu Health & Medical Center, Shizuoka 410-2315, Japan; kenken@izu-hmc.ecnet.jp; 3Kageyama Urology Clinic, Shizuoka 420-0838, Japan; kage3309@vmail.plala.or.jp

**Keywords:** saw palmetto extract, women, urinary symptoms

## Abstract

Saw palmetto berry extract (SPE) is the most commonly consumed supplement by men with benign prostatic hyperplasia (BPH). The oral administration of SPE was previously shown to significantly attenuate urodynamic symptoms in the hyperactive bladders of female rats by increasing bladder capacity and prolonging the micturition interval. The amelioration of urodynamic symptoms by SPE may be partly attributed to its binding to muscarinic receptors in the urinary bladder and its inhibition of vanilloid receptors on afferent nerves. Therefore, SPE may be pharmacologically effective at mitigating lower urinary tract symptoms (LUTS) in women. The efficacy and safety of a 12-week treatment with SPE in adult women with urinary symptoms were examined herein. The daytime frequency score in the core lower urinary symptom score (CLSS) questionnaire was significantly lower in women with LUTS treated with SPE for 12 weeks than in the placebo group. A subgroup analysis revealed that SPE alleviated the symptoms of daytime frequency (CLSS Q1) and nocturia (CLSS Q2) in a subset of subjects with a CLSS Q5 score of 1 or higher. The daytime frequency of urination in overactive bladder symptom score (OABSS) Q1 was also significantly improved by the SPE treatment. In conclusion, the present study is the first to demonstrate the potential of SPE to mitigate LUTS in adult women.

## 1. Introduction

Lower urinary tract symptoms (LUTS) due to benign prostatic hyperplasia (BPH) in men and overactive bladder (OAB) in men and women are common disorders in elderly subjects. Typical symptoms include an increased frequency of urination, nocturia, urgency, hesitancy, and a weak urine stream. Medical therapies to treat LUTS include α_1_-blockers, 5α-reductase inhibitors, antimuscarinic agents, phosphodiesterase 5 inhibitors, β_3_-adrenoceptor agonists and phytotherapy, several of which may be used in combination [1,2]. Medications with antimuscarinic effects are prescribed to elderly individuals who frequently develop anticholinergic adverse events in the peripheral and central tissues such as dry mouth, constipation, dry eyes, cognitive dysfunction, confusion and falls [3,4,5]. 

Herbs have been extensively used in Europe as natural medicines, with 50% of German urologists preferring to prescribe plant-based extracts over synthetic drugs. Saw palmetto berry extract (SPE) is one of the most commonly consumed supplements by men with BPH, often as an alternative to pharmaceutical agents [6,7,8,9,10]. A meta-analysis of the effects of SPE on urinary disorders revealed that SPE attenuated the symptoms of frequent urination in patients with BPH [11,12]. The following mechanisms have been proposed for the pharmacological effects of SPE: the inhibition of 5α-reductase [13], antiandrogenic effects [14], and antiproliferative effects [15]. Furthermore, SPE has a_1_ adrenoceptor-inhibitory properties [16], anti-inflammatory properties [17], and spasmolytic activity [18]. We previously revealed that the oral administration of SPE significantly attenuated urodynamic symptoms in the hyperactive bladders of not only male, but also female rats by increasing bladder capacity and prolonging the micturition interval [9,19,20,21,22]. The amelioration of urodynamic symptoms by SPE may be partly attributed to its binding to pharmacologically relevant receptors in the urinary bladder [9,19,20]. SPE was also recently suggested to inhibit the function of vanilloid receptors on bladder afferent nerves that transmit the sensation of the desire to void to the brain [23].

Collectively, these findings suggest that SPE is pharmacologically effective at mitigating LUTS not only in men with BPH, but also in women with OAB. However, limited information is currently available on the effects of SPE on urinary disorders in women. Despite many elderly women currently being affected by urinary disorders, there are few health food products that are specifically effective for women. If effective, SPE may improve the quality of life (QOL) of women with urinary symptoms. The present study aimed to examine the efficacy and safety of the repeated consumption of SPE in adult women with urinary symptoms.

## 2. Materials and Methods

### 2.1. Subjects

Seventy-six elderly subjects were recruited from Kageyama Clinic, Takido Clinic, Hagiwara Clinic, and Izu Hoken Medical Center as well as from the community (Figure 1). Female subjects older than 50 years were included in the present study. Subjects were excluded if they were suspected to have OAB according to the core lower urinary tract symptom score (CLSS) [24] and the OAB symptom score (OABSS). Subjects were selected based on inclusion and exclusion criteria after providing written consent.

### 2.2. Selection Inclusion Criteria

Inclusion criteria were as follows:The ability to provide written consent prior to participation in the study;Women with frequent urination, nocturia, and/or urgency for at least 2 months;Urinary symptoms that do not require medical treatment based on definitions by physicians;The ability to take the test product for the purpose of research during the study period;The ability to take daily notes during the study period.

### 2.3. Exclusion Criteria 

Exclusion criteria were as follows:Received treatments for urinary disorders within the past 2 months;Unable to have the desire to urinate;Have dysuria as the main symptom;Unable to communicate;Have a lifestyle-related disease;Currently enrolled in/will be enrolled in other studies;Receiving medications, newly designated quasi-drugs, Kampo medicine, health food products, and/or supplements for urinary disorders;Any other conditions considered to be inappropriate by the principal investigator.

### 2.4. Randomization and Blinding 

Subjects were randomized according to their age and CLSS to placebo and SPE treatment (active) groups using the permuted block technique. One subject withdrew from the study prior to the start date; therefore, the study was initiated with 75 subjects (Figure 1). A randomization table was kept sealed until the end of the study and was opened after the study period.

### 2.5. Study Design

This was a randomized, double-blind, placebo-controlled, parallel-group study. The study protocol was approved as a specified clinical trial by Hamamatsu University School of Medicine Research Ethics Board on 2 June 2020. The present study was subsequently registered in the Japan Registry of Clinical Trials (jRCT) on 15 June 2020 (#jRCTs041200018). The present study was performed in accordance with the Declaration of Helsinki and the Clinical Trials Act and its regulations to ensure respect for all human rights and to protect their safety and welfare. The written consent was obtained from subjects after they had been given a full explanation about the study. The present study was initiated on 15 June 2020 with subject recruitment, and the observation period ended on 30 November 2020. There were no changes to the study protocol during the study period.

### 2.6. Intervention

Subjects were administered treatments (test vs. placebo products) for 12 weeks from 17 August to 30 November 2020. Subjects in the active group were provided with 320 mg of SPE (SC BGG Japan Co., Ltd., Tokyo, Japan) embedded in gelatin capsules (*Yawata Saw Palmetto*^®^, Yonago, Japan), while those in the placebo group were provided with 320 mg of a glycerin fatty acid ester (food additive: Sunsoft No.707 (Taiyo Kagaku Co., Ltd., Tokyo, Japan) caprylic acid mono/diglyceride) embedded in gelatin capsules. Both products were provided by Yawata Corporation (Yonago, Japan). Test and placebo products were both administered once daily in a capsule for 12 weeks.

### 2.7. Evaluations

Outcomes were evaluated before and 12 weeks after the treatment. Four and 8 weeks after the treatment, we also contacted subjects by phone to collect CLSS and ensure compliance to the study protocol. Subjects were excluded from the present study if they withdrew their consent or there was a protocol deviation. Complications and adverse events were recorded if reported by subjects. If subjects had been taking any medications, newly designated quasi-drugs, Kampo medicine, health food products, and/or supplements unrelated to urinary disorders, they were requested not to change their usage or dose during the study period.

### 2.8. Primary Outcomes

The following data from CLSS before and 4, 8, and 12 weeks after the treatment were included as primary outcomes: daytime frequency, nocturia, urgency, urgency incontinence, stress incontinence, a slow stream, straining, incomplete emptying, bladder pain, urethral pain, and the total score.

### 2.9. Secondary Outcomes

The following were collected as secondary outcomes: OABSS before and 12 weeks after the treatment, the attenuation of symptoms (patient global impression (PGI)) 12 weeks after the treatment, and a health-related QOL questionnaire (SF-36v2^®^ (MOS 36-Item Short-Form Health Survey).

### 2.10. Safety

Demographic data included height, weight, blood pressure, and pulse rate. A blood examination was performed for aspartate aminotransferase (AST), alanine aminotransferase (ALT), γ-glutamyl transferase (γ-GTP), creatine phosphokinase (CPK), total protein, albumin, total cholesterol, triglycerides, blood urea nitrogen, creatinine, uric acid, and estimate glomerular filtration rate (e-GFR) (male = 194 × SCr^−1.094^ × Age^−0.287^, female = male × 0.739). A urinalysis was conducted to quantify the levels of protein, glucose, urobilinogen, bilirubin, and ketone bodies and to test for hematuria, specific gravity, and pH. Tests for demographics, blood, urine, and cognitive functions were performed before and 12 weeks after the treatment. Safety was assessed based on the results of the blood examination and demographic data, the numbers and types of adverse events, and complications.

### 2.11. Statistical Analysis

To compare the outcomes between the placebo and active groups, the primary endpoint was the change after 12 weeks of treatment analyzed by the unpaired Student’s *t*-test. The OABSS was analyzed by an analysis of variance (ANCOVA) with the baseline score as the covariate and the change after 12 weeks as the objective variable. SPSS Statistics ver. 25 was used for all statistical analyses, and *p* < 0.05 was considered to be significant.

## 3. Results

### 3.1. Study Subjects

Figure 1 shows a flow diagram of study subjects. The present study was conducted in accordance with, and without any changes to, the protocol. Three subjects were excluded from the analysis based on the exclusion criteria by their physicians. Therefore, 72 subjects who met the criteria described in the clinical study protocol (per protocol set) were included in the analysis. Study subjects in each group were categorized by age as follows: 50–59 (*n* = 5), 60–69 (*n* = 15), 70–79 (*n* = 16), and older than 80 (*n* = 2) in the SPE treatment (active) group and 50–59 (*n* = 4), 60–69 (*n* = 15), 70–79 (*n* = 16), and older than 80 (*n* = 2) in the placebo group. Table 1 shows baseline characteristics in the active and placebo groups.

### 3.2. Primary Efficacy Outcomes

Table 2 shows the results of CLSS, which was the primary outcome. Changes in the average score for CLSS question 1 (Q1: “How many times do you typically urinate from waking in the morning until sleeping at night” (daytime frequency)) from the baseline to post-treatment were significantly smaller in the active group (−0.8 ± 0.9) than in the placebo group (−0.4 ± 0.8) (*p* = 0.04).

As shown in Table 3, a subgroup analysis of subjects with a CLSS (Q5: stress incontinence) score of 1 or higher showed that changes in average scores from the baseline to post-treatment were significantly smaller in the active group than in the placebo group for both CLSS Q1 (−1.0 ± 0.9 vs. −0.3 ± 0.9, *p* = 0.03) and Q2 (“How many times do you typically urinate from sleeping at night until waking in the morning”) (nocturia) (−0.7 ± 0.7 vs. −0.3 ± 0.5, *p* = 0.04).

### 3.3. Secondary Efficacy Outcomes

Table 4 summarizes the outcomes of OABSS. Similar to CLSS, changes in the average score for OABSS question 1 (Q1: “How many times do you typically urinate from waking in the morning until sleeping at night” (daytime frequency)) from the baseline to post-treatment were significantly smaller in the active group (−0.4 ± 0.6) than in the placebo group (−0.1 ± 0.7) (*p* = 0.04). No significant differences were observed in (Patient Global Impression) PGI or SF-36v2^®^ (MOS 36-Item Short-Form Health Survey) between the groups.

### 3.4. Safety

No significant differences were noted in demographics or blood and urine test results before and after the treatment between the SPE treatment (active) and placebo groups (Table 5). There were no adverse events related to the treatment.

## 4. Discussion 

Natural medicines have been widely used for thousands of years in many regions of the world, including China, India, Egypt and Greece. These medicines have recently gained popularity in many Western countries due to their reliable therapeutic effects and affordability. With the increasingly extensive application of natural products and their bioactive ingredients to disease treatment, the therapeutic responses of these products need to be clearly identified to ensure their therapeutic efficacy and safety [25]. SPE is a phytotherapeutic agent that is the most commonly used to treat BPH and LUTS in men and has been the most thoroughly examined [9,10]. Our previous findings from preclinical studies indicated that SPE might effectively mitigate LUTS in women with OAB [22].

Homma et al. [24] developed the CLSS questionnaire for non-disease-specific symptoms to readily address 10 important LUTS, which allowed for a simple and comprehensive assessment of female LUTS. The CLSS questionnaire assesses 10 urination-related symptoms, each scored on a scale from 0 to 3 [26]. This simple questionnaire is applicable to both men and women with a wide variety of urinary impairments [24,27,28]. Fujiwara et al. [27] reported that the CLSS questionnaire was more comprehensive than the international prostate symptom score (IPSS) questionnaire for symptom assessment, based on a comparison of LUTS evaluations performed with IPSS versus CLSS. Therefore, CLSS is widely used to assess LUTS. The CLSS questionnaire enables a simple and comprehensive assessment of female LUTS [27]. The present study used the CLSS questionnaire and demonstrated that the daytime frequency score (CLSS Q1) after the 12-week treatment with SPE was significantly lower in the active group than in the placebo group. A subgroup analysis revealed that SPE alleviated the symptoms of daytime frequency (CLSS Q1) and nocturia (CLSS Q2) in a subset of subjects with a CLSS Q5 of 1 or higher. The daytime frequency of urination in OABSS Q1 was also significantly improved by the SPE treatment compared with the placebo. To the best of our knowledge, this is the first study to demonstrate that SPE attenuated urinary symptoms in adult women. The observed mitigation of LUTS by SPE might be more prominent in female patients with OAB who were excluded in the present study.

The findings of our recent cystometry study showed that the oral administration of SPE significantly attenuated frequent urination in female rats by prolonging voiding intervals and increasing voided volumes, with antagonistic effects on the acetylcholine-induced contraction of bladder smooth muscle and with muscarinic receptor binding activity [22]. Similar effects to those of SPE were observed for antimuscarinic agents, which attenuate LUTS in patients with OAB mainly by antagonizing bladder muscarinic receptors, as reviewed by Yamada et al. [5]. Therefore, the amelioration of frequent urination by SPE may be partly attributed to the relaxation of bladder smooth muscle through the antagonization of muscarinic receptors in the bladder. Furthermore, a recent study showed that SPE bound to and suppressed the function of transient receptor potential vanilloid subtype 1 (TRPV1) [23], which is involved in afferent nerve function [29]. Collectively, these findings suggest that SPE increased urine storage function by relaxing bladder smooth muscles in female rats through the blockade of muscarinic receptors, and also by suppressing the afferent nerve function of the bladder. The pharmacological effects of SPE reported in these preclinical studies [22,23] may have contributed to the mitigation of daytime frequency and nocturia by the repeated oral intake of SPE for 12 weeks in adult women with urological symptoms. 

Previous studies showed the significant mitigation of BPH and LUTS by SPE supplementation [7,11,12,30], whereas others found no benefits [31,32]. These discrepancies may be the result of differences in the putative active components, fatty acids and phytosterol, of SPE supplements [33,34,35]. Multiple effects derived from different constituents, such as fatty acids and phytosterol, may contribute to SPE efficacy [9,36]. Further studies on the clinical relevance of these bioactive components are needed for the standardization of SPE, which will enable consistent efficacy and recommendations for use in the treatment of LUTS. SPE is mainly composed of 90% saturated and unsaturated fatty acids as well as higher alcohols and sterols [9,33]. Therefore, the pharmacological effects of SPE may be attributed to free fatty acids in SPE, as revealed by Abe et al. [37,38]. They isolated free fatty acids (e.g., oleic acid and lauric acid) from SPE and showed the significant binding of these fatty acids to pharmacological receptors, such as muscarinic, α_1_-adrenergic and 1,4-dihydropyridine calcium channel antagonist receptors. The safety of SPE in women in the present study was demonstrated by the lack of significant differences in demographics, blood and urine test results. Furthermore, the safety of SPE in humans, including Japanese subjects, has already been confirmed following the intake of a clinical dose [8,39], excess dose [40], and the 18-month intake of an excess dose [41]. 

Overall, the present study demonstrated for the first time that SPE may mitigate the symptoms of daytime frequency and nocturia in women. The results obtained suggest the potential of SPE to mitigate LUTS in women.

## Figures and Tables

**Figure 1 nutrients-14-01190-f001:**
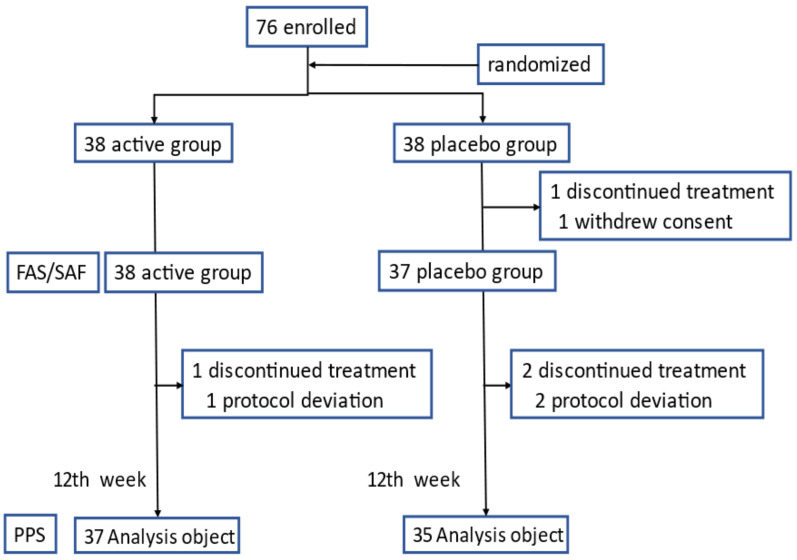
Flow diagram of participants throughout the study. FAS/SAF: Full Analysis Set/Safety Analysis Set; PPS: Per Protocol Set.

**Table 1 nutrients-14-01190-t001:** Subject backgrounds in SPE treatment (active) and placebo groups.

Variable	Active Group (*n* = 38)		Placebo Group (*n* = 37)		*p* Value
	Mean	SD	Mean	SD	
Age (years)	69.3	7.0	69.6	6.4	0.84
Blood pressure (mmHg)					
Systolic	135.4	21.0	134.7	21.8	0.88
Diastolic	74.7	13.4	71.5	12.1	0.29
CLSS:					
Daytime frequency	1.1	0.8	1.0	0.7	0.47
Nocturia	1.2	0.6	1.0	0.7	0.30
Urgency	0.4	0.6	0.5	0.6	0.76
Urgency incontinence	0.3	0.6	0.2	0.4	0.54
Stress incontinence	0.8	0.7	0.7	0.6	0.56
Slow stream	0.6	0.7	0.6	0.8	0.58
Straining	0.1	0.4	0.4	0.7	0.11
Incomplete emptying	0.2	0.5	0.2	0.5	0.96
Bladder pain	0.0	0.0	0.1	0.4	0.18
Urethral pain	0.0	0.0	0.1	0.4	0.18
Total score	4.7	2.0	4.7	3.1	0.91
OABSS:					
Daytime frequency	0.7	0.5	0.7	0.5	0.94
Nocturia	1.2	0.7	1.1	0.7	0.36
Urgency	0.4	0.5	0.5	0.5	0.74
Urgency incontinence	0.3	0.6	0.2	0.5	0.71
Total score	2.6	1.3	2.5	1.4	0.58
Total protein (g/dL)	7.0	0.3	6.9	0.4	0.58
Total bilirubin(mg/dL)	0.7	0.3	0.7	0.3	0.84
AST (U/L)	19.7	4.9	22.6	7.4	0.05
ALT (U/L)	16.3	6.3	18.6	8.8	0.20
γ-GTP (IU/L)	24.6	20.7	25.1	16.8	0.91
Total cholesterol (mg/dL)	210.3	37.3	216.7	34.8	0.44
Triglyceride(mg/dL)	151.7	78.7	152.8	95.0	0.96
eGFR (mL/min/1.73 m^2^)	66.8	11.6	68.4	15.1	0.61
CPK (U/L)	112.7	56.1	100.7	45.5	0.31
Uric acid (mg/dL)	4.8	1.1	5.0	1.0	0.52
Blood urea nitrogen(mg/dL)	15.8	4.0	17.0	4.1	0.18
Creatinine(mg/dL)	0.7	0.1	0.7	0.1	0.74

Differences between the active and placebo groups were analyzed using the unpaired Student’s *t*-test for means. CLSS: core lower urinary tract symptom score, OABSS: overactive bladder symptom score, AST: aspartate aminotransferase, ALT: alanine aminotransferase, g-GTP: gamma-glutamyl transpeptidase, eGFR: estimated glomerular filtration rate (male = 194 × SCr^−1.094^ × Age^−0.287^, female = male × 0.739), CPK: creatine phosphokinase. Values are shown as the mean and standard deviation (SD).

**Table 2 nutrients-14-01190-t002:** Mean difference from the baseline for core lower urinary tract symptom scores (CLSS) in SPE treatment (active) and placebo groups.

Variable		Active Group (*n* = 37)		Placebo Group (*n* = 35)		*p* Value
		Mean	SD	Mean	SD	
Daytime frequency	4 week	−0.4	0.9	−0.2	1.0	
	8 week	−0.4	1.1	−0.3	0.8	
	12 week	−0.8	0.9	−0.4	0.8	0.04 *
Nocturia	4 week	−0.1	0.7	−0.1	0.8	
	8 week	−0.1	0.7	−0.2	0.6	
	12 week	−0.6	0.7	−0.4	0.6	0.22
Urgency	4 week	0.0	0.6	0.0	0.7	
	8 week	−0.1	0.7	0.0	0.6	
	12 week	−0.1	0.7	0.0	0.7	0.50
Urgency	4 week	−0.1	0.7	−0.1	0.5	
incontinence	8 week	−0.2	0.6	−0.1	0.5	
	12 week	−0.2	0.6	−0.1	0.5	0.71
Stress incontinence	4 week	−0.1	0.7	−0.3	0.7	
	8 week	−0.4	0.9	−0.3	0.6	
	12 week	−0.2	0.8	−0.4	0.6	0.17
Slow stream	4 week	0.0	0.8	0.1	0.8	
	8 week	0.0	0.7	0.1	1.0	
	12 week	0.1	0.8	−0.1	0.8	0.57
Straining	4 week	0.1	0.4	0.1	0.7	
	8 week	0.0	0.4	0.1	0.6	
	12 week	0.1	0.4	0.1	0.8	0.83
Incomplete emptying	4 week	0.0	0.5	0.0	0.5	
	8 week	0.0	0.4	0.1	0.7	
	12 week	−0.1	0.5	0.1	0.7	0.23
Bladder pain	4 week	0.0	0.0	−0.1	0.4	
	8 week	0.0	0.0	−0.1	0.4	
	12 week	0.0	0.0	−0.1	0.4	0.18
Urethral pain	4 week	0.0	0.0	−0.1	0.4	
	8 week	0.0	0.0	−0.1	0.4	
	12 week	0.0	0.0	−0.1	0.4	0.15
Total score	4 week	−0.6	2.7	−0.7	3.5	
	8 week	−1.2	2.3	−0.9	3.6	
	12 week	−1.8	2.6	−1.3	3.1	0.49

The analysis was performed on data obtained before and 12 weeks after the treatment. Differences between the placebo and active groups were analyzed by the unpaired Student’s *t*-test for means. Values are shown as the mean and standard deviation (SD). * *p* < 0.05.

**Table 3 nutrients-14-01190-t003:** Mean difference from the baseline for core lower urinary tract symptom scores (CLSS) in SPE treatment (active) and placebo groups with a score of 1 or higher for CLSS Q5.

Variable		Active Group (*n* = 23)		Placebo Group (*n* = 22)		*p* Value
		Mean	SD	Mean	SD	
Daytime frequency	4 week	−0.6	0.9	−0.2	1.0	
	8 week	−0.6	1.0	−0.4	0.9	
	12 week	−1.0	0.9	−0.3	0.9	0.03 *
Nocturia	4 week	−0.1	0.7	0.0	0.7	
	8 week	−0.2	0.7	−0.1	0.6	
	12 week	−0.7	0.7	−0.3	0.5	0.04 *
Urgency	4 week	0.0	0.6	−0.2	0.7	
	8 week	−0.1	0.7	0.0	0.7	
	12 week	−0.1	0.6	0.0	0.8	0.53
Urgency incontinence	4 week	−0.1	0.7	−0.2	0.6	
	8 week	−0.1	0.6	−0.2	0.6	
	12 week	−0.2	0.5	−0.1	0.6	0.64
Stress incontinence	4 week	−0.4	0.5	−0.5	0.7	
	8 week	−0.9	0.7	−0.6	0.6	
	12 week	−0.5	0.7	−0.7	0.6	0.39
Slow stream	4 week	0.0	0.6	0.0	0.9	
	8 week	0.1	0.8	0.0	1.0	
	12 week	0.1	0.7	0.0	0.9	0.87
Straining	4 week	0.1	0.4	0.0	0.7	
	8 week	0.0	0.3	0.1	0.8	
	12 week	0.1	0.5	0.1	0.9	0.98
Incomplete emptying	4 week	0.1	0.5	0.0	0.6	
	8 week	0.1	0.5	0.1	0.8	
	12 week	0.0	0.4	0.1	0.8	0.47
Bladder pain	4 week	0.0	0.0	−0.1	0.5	
	8 week	0.0	0.0	−0.1	0.5	
	12 week	0.0	0.0	−0.1	0.5	0.19
Urethral pain	4 week	0.0	0.0	−0.1	0.5	
	8 week	0.0	0.0	−0.1	0.5	
	12 week	0.0	0.0	−0.1	0.5	0.19
Total score	4 week	−1.0	2.3	−1.3	3.8	
	8 week	−1.6	2.1	−1.3	4.2	
	12 week	−2.3	2.3	−1.4	3.6	0.32

The analysis was performed on data obtained before and 12 weeks after the treatment. Differences between the placebo and active groups were analyzed by the unpaired Student’s *t*-test for means. Values are shown as the mean and standard deviation (SD). * *p* < 0.05.

**Table 4 nutrients-14-01190-t004:** Mean difference from the baseline for the overactive bladder symptom score (OABSS) in SPE treatment (active) and placebo groups.

Variable	Active Group (*n* = 37)		Placebo Group (*n* = 35)		*p* Value
	Mean	SD	Mean	SD	
Daytime frequency	−0.4	0.6	−0.1	0.7	0.04 *
Nocturia	−0.6	0.9	−0.4	0.7	0.68
Urgency	0.0	0.8	0.1	0.7	0.46
Urgency incontinence	−0.1	0.8	−0.1	0.5	0.56
Total score	−0.6	2.3	−0.6	1.3	0.76

The analysis was performed on data obtained before and 12 weeks after the treatment. Differences between the placebo and active groups were examined by an analysis of covariance (ANCOVA) for means. Values are shown as the mean and standard deviation (SD). * *p* < 0.05.

**Table 5 nutrients-14-01190-t005:** Mean difference for demographics or blood and urine test results from the baseline and 12-week treatment with SPE (active) and placebo.

Variable		Baseline		12-Week Treatment	
		Mean	SD	Mean	SD
Weight (kg)	Active	53.5	7.4	53.8	7.2
	Placebo	55.2	9.7	55.5	9.9
Blood pressure (mmHg)					
Systolic	Active	135.6	21.3	135.1	21.7
	Placebo	134.7	21.8	133.0	19.8
Diastolic	Active	74.8	13.6	75.2	12.4
	Placebo	71.5	12.1	73.0	9.2
Pulse	Active	70.6	8.8	73.5	11.7
	Placebo	70.2	11.4	70.6	11.5
Total protein (g/dL)	Active	7.0	0.3	7.1	0.3
	Placebo	6.9	0.4	7.0	0.3
Total bilirubin(mg/dL)	Active	0.7	0.3	0.6	0.2
	Placebo	0.7	0.3	0.6	0.3
AST (U/L)	Active	19.5	4.8	21.2	6.6
	Placebo	22.6	7.4	22.8	5.8
ALT (U/L)	Active	16.1	6.3	18.4	10.2
	Placebo	18.6	8.8	19.7	9.9
γ-GTP (IU/L)	Active	24.1	20.7	23.0	20.2
	Placebo	25.1	16.8	23.0	12.1
Total cholesterol (mg/dL)	Active	209.4	37.4	218.9	35.9
	Placebo	216.7	34.8	219.3	36.0
Triglyceride(mg/dL)	Active	149.7	78.8	145.5	91.0
	Placebo	152.8	95.0	128.9	83.8
eGFR (mL/min/1.73 m^2^)	Active	66.1	10.8	68.6	12.1
	Placebo	68.4	15.1	70.1	14.5
CPK (U/L)	Active	114.6	55.6	120.4	96.9
	Placebo	100.7	45.5	104.6	45.5
Uric acid (mg/dL)	Active	4.8	1.1	4.9	1.3
	Placebo	5.0	1.0	4.9	1.1
Blood urea nitrogen(mg/dL)	Active	15.7	4.0	15.7	3.1
	Placebo	17.0	4.1	15.6	4.2
Creatinine(mg/dL)	Active	0.7	0.1	0.7	0.1
	Placebo	0.7	0.1	0.7	0.1

Active group (*n* = 37), Placebo group (*n* = 37). AST: aspartate aminotransferase, ALT: alanine aminotransferase, g-GTP: gamma-glutamyl transpeptidase, eGFR: estimated glomerular filtration rate (male = 194 × SCr^−1.094^ × Age^−0.287^, female = male × 0.739), CPK: creatine phosphokinase. Values are shown as the mean and standard deviation (SD).

## Data Availability

Data outputs for this study are available by contacting the corresponding author.

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
