# Peer review of "Beneficial Effects of Saw Palmetto Fruit Extract on Urinary Symptoms in Japanese Female Subjects by a Multicenter, Randomized, Double-Blind, Placebo-Controlled Study"

_nutrients, 2022, doi:10.3390/nu14061190_

Round 1

Reviewer 1 Report

Dear authors,

First, I would like to congratulate the team for the effort. It is nos easy to perform an independent double-blind clinical trial. I believe that, independently of the results (positive or negative), clinical trials should always be published.

Nevertheless, I have concerns about how the data has been analysed and interpreted, as well as some suggestions:

General comments:

  • For a clinical trial, I am always fond of uploading the protocol as supplementary material.
  • Authors should also upload as supplementary material the questionnaires used in the study.
  • Authors agree that they have received funding from Yawata coorporation. What is the role of the funder in this study?

Data and interpretation concerns:

  • Authors have used means and parametric tests. However, as far as I know, the questionnaires are qualitative (a scale that then is transformed into 1-4 scale). Thus, a mean of these values is not an appropriate measure; and ANCOVA tests are not valid in this scenario. Authors should use percentages instead. For example, they can report whether there are statistical differences in the percentage of improvement between both groups.
  • Even with the ANCOVA test, authors have only discussed those values that have been statistically significant. However, more than 60 p-values are reported within the text. Given this scenario, authors must use a multiple-test correction in order to adjust their p-values (a p-value threshold of 0.05 means that 1 out of 20 test will be statistically significant just by chance). For more info, please see this instructive article:

Sainani, K et al. Bonferroni, Holm, and Hochberg corrections: fun names, serious changes to p values. PM R 2014 Jun;6(6):544-6. doi: 10.1016/j.pmrj.2014.04.006. Epub 2014 Apr 22.

            In summary, I believe that the study is publishable, but it has to be changed. The statistics is not correct given the nature of the data, and the interpretation of the data, without adjusting for multiple comparisons, are also not valid. I would like to encourage the authors to improve the analysis plan and interpret the data accordingly, because, as I stated before, I believe that the results from a clinical trials should always be published.

Author Response

Dear Reviewer,

Please kindly see the attached file.

Thank you.

Reviewer 2 Report

Yamada et al. have carried out a clinical trial with women suffering from urinary symptoms in which they have evaluated whether the administration of saw palmetto fruit extract orally for 12 weeks is capable of improving these symptoms.
The introduction of the article is very complete and well written, as well as the discussion of the results. However, I think it is necessary to improve some aspects of the methodology and the results before the definitive publication of the article:
- In the title, I would change the ; which is after "subjects" by :
- The last four lines of the introduction talk about the methodology and the results. They should be removed from this section, as this information appears again later in its corresponding section.
- Why does "Experimental Section" appear in parentheses in the title "Materials and methods"?
- There is repeated information in the "Subjects", "Selection inclusion criteria", "Exclusion criteria", "Randomization and blinding" and "Study design" subsections. I believe that these sections should be reviewed and all the information that is repeated should be eliminated, since it is not necessary for it to appear several times.
- In section 2.7. it states "If subjects had been taking any medications, newly designated quasi-drugs, Kampo medicine, health food products, and/or supplements un-related to urinary disorders, they were requested not to change their usage or dose during the study period. ". I do not understand this, because in the exclusion criteria, the consumption of these medications was an absolute reason for exclusion.
- Section 2.10. there are many abbreviations not described.
- Regarding the statistical analysis, I believe that it has not been carried out correctly and that it should be repeated. In the first place, it is not specified whether the normality of the data has been evaluated or with which test, so performing a posterior ANCOVA or Student's t-test may be erroneous. Secondly, it is not correct to use an unpaired Student's t-test when what is being compared are the values of the same group of individuals at different times. A paired Student's t test should be used. The same problem exists when it is indicated that an ANCOVA has been used to compare between the two study groups. It does not make sense, because when we work with two different groups of individuals, the unpaired Student's t-test must be used. 

- Again, in section 3.1. information appears that has already been described in the methodology. Avoid repeating information.
- Section 3.4: Where are the results discussed in this section presented? I think they should also be presented, as they are very important to confirm the safety of the treatment under evaluation.
- Tables and figures: I think it is more correct to call it p-value. Also, it is necessary to define in the description of the table or figure all the abbreviations that are presented.
- Table 1: What equation has been used to calculate the eGFR? It must be indicated.

Author Response

(The authors gave the same response as above.)

Reviewer 3 Report

To the Authors

Thank you for the opportunity of revising this study. I enjoyed reading the study. However, several flaws deserve attention and are therefore listed below.

Abstract:

Line 9: please replace “administered” with “treated with”

*****

Last line:

Is this the first study in female patients?

Maybe the authors could point out that SPE was investigated in male BPH patients and not female patients, thus the rational for this study.

**********

Introduction

First two lines:

BPH is not a disorder in women and men ...please rephrase these two lines.

*****

Last two lines:

“The results obtained in the present study indicate the po-tential of SPE to mitigate storage symptoms in women.”

Remove this sentence from the introduction - it is supposed to be in the conclusion section.

**********

Selection:

Is there a connection - on which the authors based their choice- between female rats and the choice of only female subjects – instead of male and female subjects?

*****

Inclusion criteria:

  • “Urinary symptoms that do not require medical treatment based on definitions by physicians”

How did the physicians decide? according to the subjective complaints of the paient?

*****

Exclusion Criteria:

“Unable to have the desire to urinate”

Do the authors mean detrusor inactivity?

**********

Results

3.2. Primary efficacy outcomes

“Changes in the average score for CLSS question 5 (Q5: “Leaking of urine when you cough, sneeze, or strain” (stress incontinence)) from the baseline to post-treatment were significantly smaller in the placebo group (-0.4±0.6) than in the SPE treatment (active) group (-0.2±0.8) (p=0.048).”

The author maybe mean greater not smaller? – there is a contradiction with what is written in the abstract concerning CLSS Q5

*****

Why would SUI - an anatomic problem- be treated with SPE or expected to get better with such treatment?

how would the authors explain the changes of SPE to an anatomic problem like SUI?

**********

Discussion

Limitations? of the study

*****

I would suggest adding a comment that although a benefit was demonstrated based on questionnaires, true benefit can only be demonstrated based on urodynamic studies of subjects.

Thank you,

Author Response

(The authors gave the same response as above.)

Round 2

Reviewer 2 Report

The authors have applied all suggested changes successfully.